# Analysis of Voice, Speech, and Language Biomarkers of Parkinson’s Disease Collected in a Mixed Reality Setting

**DOI:** 10.3390/s25082405

**Published:** 2025-04-10

**Authors:** Milosz Dudek, Daria Hemmerling, Marta Kaczmarska, Joanna Stepien, Mateusz Daniol, Marek Wodzinski, Magdalena Wojcik-Pedziwiatr

**Affiliations:** 1Department of Measurement and Electronics, AGH University of Krakow, 30-059 Krakow, Poland; hemmer@agh.edu.pl (D.H.); kaczmarskamarta2@gmail.com (M.K.); joaste@student.agh.edu.pl (J.S.); daniol@agh.edu.pl (M.D.); wodzinski@agh.edu.pl (M.W.); 2Department of Neurology, Andrzej Frycz Modrzewski Krakow University, 30-705 Krakow, Poland; mwp283@icloud.com

**Keywords:** explainable artificial intelligence, large language models, mixed reality, Parkinson’s disease, voice biomarkers, remote patient monitoring

## Abstract

This study explores an innovative approach to early Parkinson’s disease (PD) detection by analyzing speech data collected using a mixed reality (MR) system. A total of 57 Polish participants, including PD patients and healthy controls, performed five speech tasks while using an MR head-mounted display (HMD). Speech data were recorded and analyzed to extract acoustic and linguistic features, which were then evaluated using machine learning models, including logistic regression, support vector machines (SVMs), random forests, AdaBoost, and XGBoost. The XGBoost model achieved the best performance, with an F1-score of 0.90 ± 0.05 in the story-retelling task. Key features such as MFCCs (mel-frequency cepstral coefficients), spectral characteristics, RASTA-filtered auditory spectrum, and local shimmer were identified as significant in detecting PD-related speech alterations. Additionally, state-of-the-art deep learning models (wav2vec2, HuBERT, and WavLM) were fine-tuned for PD detection. HuBERT achieved the highest performance, with an F1-score of 0.94 ± 0.04 in the diadochokinetic task, demonstrating the potential of deep learning to capture complex speech patterns linked to neurodegenerative diseases. This study highlights the effectiveness of combining MR technology for speech data collection with advanced machine learning (ML) and deep learning (DL) techniques, offering a non-invasive and high-precision approach to PD diagnosis. The findings hold promise for broader clinical applications, advancing the diagnostic landscape for neurodegenerative disorders.

## 1. Introduction

Neurodegenerative diseases are characterized by the progressive degeneration of neurons, leading to a gradual decline in motor and cognitive functions. Parkinson’s disease, the second most common neurodegenerative disorder after Alzheimer’s disease, exemplifies this degeneration. It results from the deterioration of dopamine-producing neurons in the brain, causing tremors, muscle rigidity, bradykinesia, and issues with balance, coordination, and speech [1]. Focusing on Parkinson’s disease’s impact on speech, voice, and language, significant vocal tract changes are observable [2]. These include dysphonia, monotony, reduced speech clarity, and increased frequency of speech interruptions [3]. Parkinson’s patients often exhibit weakened or incomplete vocal fold closure, leading to modulation difficulties. These changes stem from motor control disruptions affecting the laryngeal muscles and vocal tract structures [2,3,4].

Biomarkers are used for diagnosing and monitoring disease progression [5,6]. These biological traits can be measured and objectively evaluated as indicators of biological processes. Vocal biomarkers encompass specific acoustic and linguistic features that provide valuable insights into a patient’s health status [7,8]. Their analysis involves both acoustic parameters such as tone, tempo, volume, and modulation as well as linguistic characteristics, including word choice, syntax, and fluency [8]. Due to this fact, those biomarkers are utilized not only for diagnosing and monitoring diseases but also for predicting disease progression and evaluating therapy effectiveness [5,6]. In the case of Parkinson’s disease, these voice disorders typically originate with subtle changes in phonation and articulation but progressively worsen over time. Initially, slight pitch instability, significantly quieter speech (mild reductions in energy within the higher harmonic spectrum), monotonicity, and slowness may appear, which is challenging to detect by the human ear [2,3,9,10]. As the condition advances, these alterations become more pronounced, leading to significant pitch fluctuations, a marked weakening of high-frequency components, and increasing imprecision in vowel and consonant articulation [11]. Additionally, linguistically, Parkinson’s patients may struggle with appropriate vocabulary selection, sentence complexity, and eloquence, tending to use more general words and shorter sentences over the disease progression [7,8,9,10]. Ultimately, these cumulative effects result in a substantial decline in speech intelligibility, making verbal communication increasingly challenging. Consequently, the growing recognition of these progressive vocal changes has driven research toward the development of reliable, non-invasive vocal biomarkers, offering the advantages of being easily obtainable, globally accessible, and cost-efficient for diagnostic purposes.

Changes in acoustic parameters and linguistic features can reflect a patient’s neurological state, with vocal biomarker research focusing primarily on Parkinson’s disease [12]. Currently, there is no standardized voice recording protocol for identifying these biomarkers, but human vocal tasks can be categorized into two main groups: verbal tasks (e.g., speaking single words, repeating short sentences, or reading text passages) and vowel/syllable tasks (e.g., sustained vowel phonation or diadochokinetic tasks) [7]. Researchers debate the efficacy of using defined single words or text read aloud at a specified pace versus allowing spontaneous speech [13,14]. A hybrid approach, where tasks are controlled through patient instructions but allow word choice for naturalness, is favored. Examples include picture descriptions or storytelling, enabling analysis of linguistic structure and communication style [8]. Sustained vowel phonations, another common recording type, involve participants maintaining a vowel sound as long and steadily as possible. These tasks provide information to assess dysphonia and estimate patient voice without articulatory, speech rate, intonation, dialect, or language variables. Syllable tasks, like rapid syllable repetition (e.g., /pe/-/ta/-/ka/), are often used to evaluate articulatory disorders, speech rate, and intelligibility [7,9].

Numerous studies have explored the potential of voice analysis for early detection of Parkinson’s disease, demonstrating significant advancements in both methodology and accuracy. A pivotal study involving 115 Parkinson’s patients and a control group utilized support vector machine (SVM) algorithms trained on voice recordings. This approach achieved high accuracy in distinguishing healthy individuals from those in the early and moderate stages of the disease, showcasing the feasibility of machine learning in this domain [15]. Similarly, research utilizing the UC Irvine (UCI) dataset highlighted the effectiveness of ensemble methods such as XGBoost and AdaBoost, with models like LightGBM achieving up to 96% accuracy in classifying Parkinson’s disease based on voice features [16]. Further evidence of the potential of voice analysis comes from studies investigating specific speech characteristics. For instance, jitter, shimmer, and Harmonics-to-Noise Ratio (HNR) were identified as significant indicators of Parkinson’s disease, with SVM models achieving 87% accuracy in classification [17]. Expanding this work to linguistic diversity, an analysis of Spanish speech corpora revealed that certain acoustic features, particularly in vowel “a”, could distinguish Parkinson’s patients from healthy controls with an accuracy of 91.3% [18].

Building on these foundations, researchers have increasingly turned to advanced machine learning techniques, such as Deep Neural Networks (DNNs). A comparative analysis using the UCI Parkinson dataset found that DNNs, which simulate the brain’s structure and processing, outperformed classical machine learning methods in accuracy, further validating their suitability for this task [19]. This trend towards deep learning (DL) is also evident in mobile device applications. Utilizing the Mobile Device Voice Recordings at King’s College London (MDVR-KCL) dataset, researchers demonstrated the feasibility of mobile devices for non-invasive detection, achieving an F1-score of 92% with K-Nearest Neighbors (KNN) and Artificial Neural Network (ANN) models [20]. Importantly, the F1-score, a key metric in machine learning, combines precision and recall into a single measure, making it particularly valuable in evaluating diagnostic tools in imbalanced datasets, such as those involving neurodegenerative diseases.

In recent years, new architectures of DL models emerged and extended the capabilities of general ANNs by leveraging deeper architectures, enabling the extraction of complex and abstract patterns from raw data. For example, a study combining Convolutional Neural Networks (CNNs) with Long Short-Term Memory (LSTM) networks achieved an F1-score of 96% by analyzing mel-spectrograms of speech recordings, demonstrating the advantages of hybrid models in capturing intricate vocal features [21]. Similarly, a standalone LSTM model achieved 93% accuracy in classifying Parkinson’s disease using spontaneous speech from a Spanish dataset [22]. Highlighting the versatility of deep learning, another study employed YAMNet for voice classification, achieving promising results with an F1-score of 0.84 and a precision of 1.0 on spontaneous speech [23].

Recent advances in artificial intelligence have introduced Large Language Models (LLMs), a class of deep learning architectures designed to process and understand large-scale text and audio data. Models such as wav2vec2, Whisper, WavLM, and HuBERT leverage self-supervised learning to pretrain on vast amounts of unannotated data, enabling them to capture complex patterns and relationships within the input. Those models have been tested for detecting irregularities in voice patterns, achieving F1-scores ranging from 75% to 83%, depending on the task [24]. Additionally, wav2vec2 has been used to extract embeddings (representations of voice features) from audio recordings, which were subsequently analyzed using classification algorithms like Random Forest and XGBoost, achieving an area under the curve (AUC) score of 0.98 on an Italian dataset involving text reading tasks [25].

Collectively, these findings underscore the transformative potential of combining advanced speech processing techniques with machine learning and deep learning models. From classical methods to sophisticated architectures like CNNs, LSTMs, and LLMs, the evolution of these approaches demonstrates the growing precision and versatility of voice analysis in non-invasive diagnostics for Parkinson’s disease. Vocal biomarkers play a crucial role in diagnosing and monitoring neurodegenerative diseases. Therefore, research communities increasingly focus on developing acoustic and linguistic analysis techniques to use subtle voice changes to help diagnose disease before progression.

While purely audio-focused approaches have achieved remarkable progress in Parkinson’s disease diagnostics, emerging technologies offer opportunities for more comprehensive evaluations. In particular, mixed reality (MR) systems provide multi-modal data capture, encompassing gaze, hand, and head movements, while simultaneously recording voice input. Head-mounted displays (HMDs) featuring multiple sensors and consistently placed microphones can bolster both early detection and rehabilitation efforts by delivering highly detailed motion and speech data. This hands-free, immersive format also enhances patient engagement, which is crucial for sustaining therapeutic benefits. Although much existing research addresses Alzheimer’s detection via augmented reality [26], MR applications for Parkinson’s disease are increasingly recognized for enabling timely interventions [27]. The sensor arrays in HMDs can detect subtle fluctuations in motor and speech function [28], offering a more nuanced perspective on disease progression compared to conventional clinical assessments. Additionally, interactive holographic exercises promise higher adherence to treatment plans and superior rehabilitation outcomes [29], while ongoing inquiries into multilingual speech biomarkers highlight the global applicability of these MR-based strategies [30]. Future innovations in device hardware, data analytics, and immersive modalities stand to minimize clinical burdens and deliver a patient-centric approach to Parkinson’s care.

In this paper, we present an algorithm that analyzes speech data collected using a data acquisition protocol with mixed reality technology. The MR system was employed to record voice, speech, and language tasks in a non-invasive manner, offering stable microphone distance and significant advantages over traditional methods, including the potential for future integration of additional modalities, such as hand movements, eye tracking, and full-body motion. The major contributions of this paper include the development of a specialized medical processing pipeline to extract significant vocal biomarkers from various tasks and the implementation of machine learning and deep learning models for automatic detection and differentiation of patients. Based on our results, we identify the speech task with the greatest potential for Parkinson’s disease detection using MR technology voice recordings and explain which acoustic parameters are significant in distinguishing between healthy control groups and individuals with such a neurodegenerative disease. This study serves as a foundational step toward demonstrating the broader applicability of MR systems in multimodal neurodegenerative disease diagnostics.

## 2. Materials and Methods

The development of an algorithm for determining whether a person has Parkinson’s disease consisted of several stages. Initially, the speech signals of the patients were processed to remove the echo present in some of them, which resulted from the environment in which the examination was conducted. In the given study, two types of data processing approaches were used—machine learning and deep learning. Further processing of the recordings depended on the method. In the machine learning approach unique, clinically significant features were extracted using feature extraction to obtain acoustic and linguistic information depending on the nature of the task. In the next step, the subset of processed data was reduced through feature selection to retain only those features that best differentiated the research groups. The final set of features was then used for statistical analysis of the values and variability of the features, prediction of class membership using a model, and examination of which features had the main impact on the model. All steps of machine learning approach are presented in Figure 1. In the case of the deep learning approach, ordinary voice recordings that had previously undergone preprocessing were used. The techniques applied to prepare the data included audio splitting, resampling, and padding. Additionally, the models implemented for classification were previously fine-tuned to recognize voice impairments. The entire workflow for deep learning methods is presented also in Figure 1.

### 2.1. Data Acquisition Protocol

For the purposes of this study, a structured sequence of speech tasks was devised to capture detailed vocal performance and potential biomarkers in participants. In addition to collecting audio data, the system was designed to record head movements, gaze patterns, and hand motion acceleration. Regarding the sound and speech acquisition, a Microsoft (MS) HoloLens 2 mixed reality head-mounted display (MR HMD) was used with a dedicated application developed in the Unity environment. The MS HoloLens 2 MR HMD is a portable, standalone holographic device that allows interaction with holograms in physical space. It is developed by Microsoft and runs on the Windows Mixed Reality platform. The device comprises sensors, cameras, an optical system, and software that enables gesture and voice recognition. The MS HoloLens 2 MR HMD includes five directional microphone channels. Two microphones are located at the bottom of the visor, responsible for capturing the user’s voice, while the other three are at the top of the visor, capturing surrounding sounds, including noise [31,32]. Figure 2 shows a patient testing the functionality of the MS HoloLens 2 MR HMD.

The sound samples were recorded at a sampling rate of 48 kHz and a resolution of 32 bits. The advantage of using MS HoloLens 2 HMD is the consistent distance from the speaker to the microphones, which is a crucial aspect of ensuring repeatability and comparability during the experiment. Additionally, the application allows continuous monitoring of spoken words and phrases by the user, providing a responsive and efficient data handling mechanism. This approach enabled the study participants to engage in various exercises. This architecture not only facilitates real-time data acquisition but also allows precise control over recording sessions, making it an integral part of the system’s overall functionality.

The experiment utilized the MS HoloLens 2 MR HMD to display tasks instructed by software developed in Unity. Each task performed by the participant allowed the collection of various data types, such as voice, gait, head and hand motion acceleration, and gaze pattern. The example of the voice task can be seen on Figure 3. The main focus of the analysis in this study was on voice-related tasks:

1. **Picture Description**: The task involved describing what was seen in a picture displayed by the head-mounted display.

2. **Daily Activities Description**: The task involved describing the basic activities performed by the participant during the day.

3. **Story Retelling (Memory Task)**: The task involved repeating a story remembered by the participant.

4. **Vowel Phonation (“a”, “e”, “i”, “o”, “u”)**: The task involved pronouncing vowels for at least 2 s.

5. **Word Repetition (“pataka”, “petaka”, “pakata”)**: The task involved pronouncing specific words—a diadochokinetic task.

### 2.2. Patient Data

The assessment of Parkinson’s disease (PD) requires precise metrics to accurately monitor disease progression and therapy effectiveness. Common metrics related to movement and speech include the Movement Disorder Society’s Unified Parkinson’s Disease Rating Scale (MDS-UPDRS), the Voice Handicap Index (VHI), and the Polish Version of Voice Disability Coping Questionnaire (PL-VDCQ).

MDS-UPDRS, developed by the Movement Disorder Society (MDS), is divided into four parts to assess non-motor experiences (Part I), motor experiences (Part II), motor examination (Part III), and motor complications (Part IV). Part III is particularly important as it involves a clinician’s assessment of motor functions through various tasks, rated from 0 (normal) to 4 (severe impairment) based on 18 questions like facial expressions, hand movements, and gait [33,34]. The total score for Part III ranges from 0 to 132, with higher scores reflecting more severe motor dysfunction. Although standardized severity thresholds are not officially established, research suggests the following score-based categories: scores between 0 and 32 indicate mild motor impairment, scores of 33–58 suggest moderate impairment, and scores above 58 correspond to severe motor impairment. These categories should be interpreted alongside clinical context and other assessment results to comprehensively evaluate Parkinson’s disease progression.

VHI evaluates the impact of voice disorders on patients’ quality of life. It consists of 30 statements, rated on a 5-point scale, where higher scores indicating greater perceived voice-related disability. The VHI is divided into three subscales: functional (e.g., “My voice problem causes me to miss work”), emotional (e.g., “I feel annoyed when people ask me to repeat”), and physical (e.g., “I feel strained when trying to produce voice”) [35,36].

PL-VDCQ is the Polish version of the Voice Disability Coping Questionnaire (VDCQ), assessing how patients cope with stress related to voice problems. It includes 15 statements rated on a 6-point scale from 0 (“never”) to 5 (“always”), based on interviews with dysphonia patients [37,38].

The Montreal Cognitive Assessment (MoCA) is a brief cognitive screening tool used to detect mild cognitive impairment (MCI) and early Alzheimer’s disease, with growing relevance in Parkinson’s disease due to its sensitivity to executive dysfunctions. It assesses several cognitive domains, including attention, memory, language, visuospatial ability, and executive functions. The test has a maximum score of 30, with scores ≥26 generally considered normal [39]. Scores from 18 to 25 indicate mild cognitive impairment, scores from 10 to 17 suggest moderate cognitive impairment, and scores below 10 are associated with severe impairment [40]; however, research for these severity ranges has not been fully established yet and these are more arbitrarily chosen ranges. MoCA is particularly valuable in PD populations where early cognitive changes might not be evident in routine clinical evaluations.

For the experiment, a group of 21 patients diagnosed with Parkinson’s disease and 36 healthy control (HC) individuals were recruited. The average age in the control group was 48.50 ± 13.24, while in the PD group, it was 53.50 ± 7.48. All patients in our study were evaluated in the ON state, as recorded in the Case Report Form (CRF), which includes the exact time since the last levodopa dose. Additionally, the MDS-UPDRS and MoCa assessment was conducted immediately before the speech recordings, ensuring a precise evaluation of the patients’ clinical state at the time of voice analysis. The medical assessment results for the PD group were as follows: MDS-UPDRS-III: 30.06 ± 12.18; VHI: 8.57 ± 1.06; PL-VDCQ: 15.46 ± 5.02, MoCA: 20.95 ± 4.23. Based on MDS-UPDRS-III total scores, motor function among PD patients was classified into three severity groups: 12 participants exhibited mild motor impairment (scores between 0 and 32), 9 showed moderate motor impairment (scores between 33 and 58), and no participants were classified as having severe motor impairment (scores above 58). Similarly, based on the MoCA score interpretation, cognitive profiles among PD participants were classified as follows: 5 patients presented with normal cognition (scores between 26 and 30), 12 with mild cognitive impairment (scores between 18 and 25), 4 with moderate cognitive impairment (scores between 10 and 17), and none exhibited severe cognitive impairment (scores below 10).

### 2.3. Speech and Voice Parametrization

The first crucial step, before working on voice signals, is to check the quality of the data to enable feature extraction. However, this is often impossible due to the presence of noise (e.g., in the form of echo in an empty room), which causes the obtained acoustic features to often have values that do not reflect the actual state. To eliminate echo noise from the recordings of the study participants, we used a technique called “spectral gating”. This process involves analyzing the spectrogram of the signal (and optionally the noise signal) and then estimating the noise threshold for each frequency band of the signal/noise. Based on this threshold, a mask is calculated that eliminates noise below the set frequency level. The implementation of this technique is available in the “noisereduce” Python library [41].

The OpenSMILE software [42] played a crucial role in the machine learning methodology for feature extraction from audio signals. In the context of acoustic analysis of the voice of individuals with Parkinson’s disease, key features are low-level descriptors (LLDs). These descriptors are widely used due to their ability to precisely describe energy-related, spectral, and vocal parameters in a person’s speech.

In this study, the primary features used to characterize the properties of speech signals were extracted using the Computational Paralinguistics ChallengE 2016 (ComParE 2016) feature set, provided by the OpenSMILE feature extractor. Although the ComParE feature set contains 6373 static features, only 65 main low-level descriptors (LLDs) were utilized, as shown in Table 1.

All LLD features used for data processing matched those provided by the OpenSMILE software. Consequently, acoustic information was obtained for each speech task. The result of this processing was the calculation of statistical values on specified time windows, such as mean, standard deviation (std), median, first quartile (Q1), third quartile (Q3), maximum (max), and minimum (min).

Additionally, because of the lack of formants F1–F5 in the OpenSMILE extractor, we used the “Parselmouth” library for the vowel task to extract those formants [43,44]. Formants in the context of speech signal analysis refer to specific resonance frequencies that occur during the articulation of speech sounds by humans. They provide significant information because their analysis allows determining what is being said (semantic aspect), who is speaking (individual aspect), and how it is being said (research aspect, e.g., medical). The main advantage of formants is their characteristic configuration, which can be determined as a pattern for most vowels, regardless of the speaker, the articulation rate, the emotions accompanying the speech, etc. [45]. Example visualizations of the difference between formants for PD and HC individuals are shown in Figure 4 and Figure 5.

For tasks such as picture descriptions, story retelling, and answering questions, we decided to extract the number and length of silences longer than 50 ms in the voice. Silence can provide valuable information about a patient’s health condition, especially in the context of neurodegenerative diseases such as Parkinson’s disease. This is due to the manner of speech of individuals with Parkinson’s disease. Their speech is often unclear, very quiet, and consists of short, rapidly spoken sentences [46]. To extract silence, we used the Voice Activity Detection (VAD) algorithm from the “webrtcvad” Python library [47]. The parameter related to the algorithm’s sensitivity was set as mode = 2 and the parameter related to the frame length for analysis was window_duration = 0.03. This allowed for obtaining information about the location of speech within the signal, and consequently, also the segments of silence. Furthermore, we only considered those silence intervals that lasted more than 50 ms. This was because values greater than 50 ms can be classified as a pause in speech among people with Parkinson’s disease [48]. Figure 6 and Figure 7 show how the VAD algorithm works in both groups.

Formant features from the vowel task, along with silences exceeding 50 ms, were also incorporated into the machine learning methodology. Similar to the OpenSMILE low-level descriptors (LLDs), these features were transformed into statistical measures, including mean, standard deviation (std), median, first quartile (Q1), third quartile (Q3), maximum (max), and minimum (min).

### 2.4. Language Parametrization

A second source of features for machine learning methodology was from the text itself. Linguistic information was acquired only for tasks where text extraction was possible, such as picture descriptions, story retelling, and answering questions. This was accomplished using the Azure Speech SDK [49]. Based on the narratives provided by the participants, linguistic features were extracted using the spaCy library [50], including the following:**Parts of Speech (POS)**. The analysis of parts of speech can help in detecting changes in communication patterns that may be early indicators of neurodegenerative diseases such as Alzheimer’s or Parkinson’s disease. For instance, a person with Alzheimer’s disease may start using simpler sentences and less complex grammatical structures, which can be indirectly determined through POS tagging. Additionally, this analysis can aid in monitoring the progression of the disease and the response to treatment, as well as help identify patterns and trends characteristic of individuals with neurodegenerative conditions in large datasets.The parts of speech analyzed included: adverb, numeral, punctuation, determiner, interjection, adjective, preposition, particle, conjunction, proper noun, auxiliary verb, pronoun, noun, verb, coordinating conjunction, and others.**Named Entity Recognition (NER)**. Named entity recognition is a crucial first step in the process of creating semantic representations. This enables more precise identification of specific events and relationships between participants. Consequently, it can contribute to a better understanding of the differences between patient groups, which can be pivotal in the context of neurodegenerative diseases.The text was analyzed to detect the occurrence of names of people, dates, times, geographical locations, organization names, and places.**Word frequency**. For word frequency, only four to five words per task were used, which were the same for both groups—Parkinson’s patients and the control group. This type of information can be used to track what type of words are mostly common in PD and HC group. For example, the most frequently spoken words by both groups for the task “daily activities description” were the following: “morning”, “lunch”, “wake up”, “home”.**Normalized number of different words in the text**. Here, all unique words in the text were counted, and their proportion relative to all spoken words was calculated. This kind of feature allow to determine how sophisticated a person’s nomenclature is, which can be also extended to the PD and HC group.**Sentiment analysis**. Sentiment analysis can be used to monitor the progression of neurodegenerative diseases by observing changes in the patient’s attitude during conversations. For example, a person suffering from Parkinson’s disease may experience a decline in well-being associated with disease progression, which translates into expressing more negative emotions during interactions. Furthermore, sentiment analysis opens up the possibility of personalized care, as doctors can better understand the psychological context of patients dealing with such neurodegenerative conditions.It was performed for each sentence spoken by the participant. This resulted in the number of positive, negative, and neutral sentiments in the text provided by the participant. This was achieved using a pretrained roBERTa model on a Polish language corpus [51].

It is important to note that all parts of speech, named entities, and the most frequently occurring words were normalized relative to all words present in the text obtained from the participant.

### 2.5. Machine Learning Methods

In order to classify whether the participant belongs to PD or HC group, we normalized all obtained features. The classifiers used for the classification task based on the collected data included logistic regression, SVM, random forests, AdaBoost, and XGBoost. The hyperparameters of all classifiers were tuned using the GridSearch technique.

A total of 20 classifiers were trained for five-fold cross-validation on all tasks:**Three for each description-based task** (picture description, story retelling, answering questions):−Trained separately on acoustic features, linguistic features, and a combination of both.**Six for the continuous vowel pronunciation task**:−Five trained on acoustic features for each vowel separately;−One trained on the combined features of all vowels.**Four for the diadochokinetic task of repeating specific words**:−Three trained on acoustic features for each word (“pataka”, “petaka”, “pakata”);−One trained on the combined features of all these words.**One final model** trained on all features of all described tasks.

For the random forest and XGBoost models, after the initial training, the 10 most important features were selected based on their importance and consistency across all 5 folds of the cross-validation. This was done for each of the 20 classifiers. This procedure significantly reduced the number of features involved in the classification, which resulted in much better performance.

To evaluate the performance of all the aforementioned models, three metrics were chosen: F1-score, precision, and recall. The selection of these metrics was due to the unevenness of the dataset, where the control group was larger than the group diagnosed with Parkinson’s disease.

### 2.6. Deep Learning Methods


**Methods**
In the deep learning approach, we decided to use the effective language models such as HuBERT-base [52], WavLM-base [53], and wav2vec 2.0 [54]. Each of these models was implemented in basic variant. In this version, datasets were trained on the LibriSpeech dataset. It is worth mentioning that the selected language models were originally trained on a large amount of data using the self-supervised learning method, which means that there are no traditional labels during training. This strategy allows for the creation of a universal representation of speech, making it easier to use these models for various speech processing tasks, such as speaker verification, recognition or authentication. Using this approach, with a slight modification, we could easily adapt the previously trained model to classify neurodegenerative diseases. All the above-mentioned models are equipped with a convolutional layer for feature extraction from the audio input, as well as a transformer-based encoder responsible for processing sequential data. Additionally, each model is equipped with a classification layer.
**Fine-tuning for Parkinson detection**
Before fine-tuning and classifying the data obtained with the MR head-mounted display (HMD), the model was pretrained to diagnose voice pathologies using the Saarbruecken Voice Database (SVD) [55]. This database contains voice recordings from over 2000 individuals, including patients with various voice pathologies. From this dataset, we utilized 1500 recordings with pathological speech characteristics, sampled at a frequency of 44.1 kHz, to fine-tune the model for recognizing voice pathologies commonly associated with neurodegenerative diseases like Parkinson’s disease.
**Data processing**
The data underwent preprocessing to prepare it for analysis. This included resampling the recordings to a sampling rate of 16 kHz, as required by the models trained on data at this frequency. Additionally, the length of each recording was standardized to 2 s. Longer recordings, such as those obtained from MR HMD, were split into random 2-s fragments to avoid losing valuable information. Conversely, shorter recordings were padded with silence to ensure consistency across the dataset. Recordings with poor quality or processing issues that made them unsuitable for model training were excluded from the analysis.
**Training process**
During the training process, the learning rate parameter was set to 0.0003 and the batch size was set to 64. The cross-entropy loss function was used, and the Adam optimization algorithm was applied during training. Model performance was evaluated using F1-score, precision, and recall calculated from the number of correctly and incorrectly classified data points. The selection of these metrics was made in the same manner as with the machine learning models due to the unevenness of the dataset, where the control group was larger than the group diagnosed with Parkinson’s disease. The cross-validation method was used during fine-tuning. SVD and mixed reality recordings were split into 5 folds during training.

## 3. Results

### 3.1. Statistical Analysis

We performed Shapiro–Wilk tests for normality for all the features obtained from all tasks. The results implied that not all the features were normally distributed, and therefore, we utilized a Mann–Whitney U test for further analysis. We set the significance level α to 5%. The test was performed for all subjects from both groups, i.e., HC and PD. We measured effect size with Cohen’s *d* index as well. We adhere to the following effect size intervals: very small (0.01≤d<0.2), small (0.2≤d<0.5), medium (0.5≤d<0.8), large (0.8≤d<1.2), and very large (d>1.2) [56]. Due to the abundance of parameters extracted from each task, we performed the analysis in two steps: (1) quantitative analysis of significant features for each task and significant features recurrent within every task group, i.e., monologue tasks (descriptions and story retelling), vowels, and diadochokinesis; (2) additional evaluation utilizing the results of the machine learning part of our study, in particular the review of important features.

Quantitative analysis of Mann–Whitney U test results is shown in Table 2. The highest number of significant acoustic features was identified in the story retelling task (257 features), followed by the diadochokinetic “pakata” task (219) and daily activity description task (182). The lowest numbers of significant acoustic features were observed in vowel phonation tasks: “u” (40), “e” (64), “i” (68), “a” (73), and “o” (85). The highest number of significant linguistic features was identified in daily activity description (8) and story retelling (8) tasks, while the picture description task demonstrated the least (3). Within each task group, there were significant features common for every task in that group: 113 acoustic features in monologue tasks, 9 in vowels tasks, 134 in diadochokinetic tasks, and 1 linguisitc feature in monologue tasks.

Statistical analysis of ten important features of the three top performing models is shown in Table 3, Table 4 and Table 5. Eight out of ten important features of the XGBoost model for the word “pataka” in the diadochokinetic task presented a *p*-value smaller than the chosen significance level. Four of the identified significant features demonstrated a very large effect size, three large, and one medium. For the XGBoost model for acoustic and linguistic features in the story retelling task, there are eight significant features. Two demonstrated a very large effect size, four large, and two medium. In regard to the XGBoost model for acoustic features in the story retelling task, there were nine significant features, where four features demonstrated a very large effect size, three large, and two medium.

To better understand the clinical validity of acoustic speech features, we examined their correlation with established medical assessments: MoCA and MDS-UPDRS-III. The purpose of this analysis was to identify speech-derived biomarkers that might reflect underlying motor or cognitive dysfunction in Parkinson’s disease patients.

#### 3.1.1. Correlation with MoCA (Cognitive Function)

A number of fundamental frequency (*F0final*)-based acoustic features showed strong positive correlations with MoCA scores. These correlations suggest that more stable and well-modulated voicing (as reflected by *F0final* measures) is associated with better cognitive performance. The following features were the most strongly correlated:**pakata_F0final_sma_quantile50: **r=0.7369;**pakata_F0final_sma_median: **r=0.7369;**u_F0final_sma_quantile50: **r=0.6280;**u_F0final_sma_median: **r=0.6280;**u_F0final_sma_quantile25: **r=0.6119;**pataka_F0final_sma_std: **r=0.5760;**o_F0final_sma_quantile25: **r=0.5692;**pakata_F0final_sma_mean: **r=0.5285;**petaka_F0final_sma_std: **r=0.5254.

These results highlight that prosodic features (particularly those related to pitch stability and range) during diadochokinetic tasks and vowel production may serve as proxies for cognitive integrity. The strongest correlations were observed in the “pakata” DDK task, suggesting this type of vocal performance may be especially sensitive to cognitive decline.

#### 3.1.2. Correlation with MDS-UPDRS-III (Motor Function)

Several acoustic features also showed moderate correlations with MDS-UPDRS-III scores, primarily associated with variability in pitch during sustained vowel phonation. These associations point to a motor origin of the dysprosody captured by these features, consistent with Parkinsonian speech.

**o_F0final_sma_std:**r=0.6238;**e_F0final_sma_std:**r=0.4067;**u_F0final_sma_std:**r=0.3961;**e_F0final_sma_quantile75:**r=0.2934;**a_F0final_sma_std:**r=0.2806;**u_F0final_sma_quantile75:**r=0.1613;**describe-story_F0final_sma_quantile75:**r=0.1226;**e_F0final_sma_mean:**r=0.1222.

Although these correlations are weaker than those observed with MoCA, they still suggest that certain acoustic measures—particularly pitch variability—are moderately informative of motor impairment. The vowel /o/ exhibited the highest correlation with MDS-UPDRS-III, supporting its relevance for clinical speech monitoring in PD. Overall, this correlation analysis demonstrates that selected acoustic features, especially those based on pitch distribution and variability, are sensitive to both cognitive and motor changes in Parkinson’s disease. Features from diadochokinetic tasks (e.g., “pakata”) showed stronger associations with cognitive impairment, while vowel phonation tasks were more reflective of motor dysfunction. These findings support the clinical potential of using speech biomarkers as non-invasive proxies for PD-related impairment.

### 3.2. Machine Learning Classification Results

Models that were used: logistic regression, SVM, random forests, AdaBoost, and XGBoost. It was decided that for random forests and XGBoost models, the models would first be trained on the full set of features and then retrained on a set of 10 features. These features were selected based on their importance and repeatability in all five splits performed using cross-validation. This allowed for obtaining features of the highest importance regardless of the split.

The results of the XGBoost model, which achieved the best metric values, are presented in Table 6. The selection of the best results from individual models for individual tasks was based on obtaining the highest metric while maintaining high values of other metrics.

In Table 6, the best results were bolded. It can be seen that for the model trained on the task of story retelling, both for acoustic and linguistic features, the best F1-score metric value of 0.90 ± 0.05 was obtained, where hyperparameters were as follows: ‘n_estimators’: 50, ‘max_depth’: 5, ‘colsample_bytree’: 0.1. It is also important to note that for the same task, but for the model trained only on acoustic features, the best recall metric value of 0.95 ± 0.01 was achieved. The hyperparameters for this model were as follows: ‘n_estimators’: 20, ‘max_depth’: 5, ‘colsample_bytree’: 0.7. Both of these models achieved high metric values while maintaining low standard deviation values. This aspect would indicate relative stability compared to other models. Ultimately, for the precision metric, the best value was obtained for the diadochokinetic task for the word “pataka”, where the model achieved a precision of 0.96 ± 0.08, with a low standard deviation value while maintaining a very high metric value. The hyperparameters were as follows: ‘n_estimators’: 10, ‘max_depth’: 5, ‘colsample_bytree’: 0.7. It is worth noting that the same precision result was achieved for the same diadochokinetic task for the word “petaka”, but the F1-score and recall metrics were significantly worse compared to the model trained on features from the word “pataka”.

### 3.3. Machine Learning Interpretability

In addition to the analysis of the results presented in Table 6, we also visualized the importance of individual features for the best models in each task. This allows for a better understanding of the basis on which the models made their main predictions. Figure 8, Figure 9 and Figure 10 show the 10 most important features that influenced the training of models on the full dataset in tasks where the highest metric values were obtained, namely: story retelling—acoustic features; story retelling—acoustic and linguistic features; diadochokinetic task for the word “pataka”; and the model trained on all features from all tasks. These 10 features were selected based on their importance and repeatability in all 5 splits performed by cross-validation.

Figure 8, Figure 9 and Figure 10 illustrate the dominance of acoustic features during model training, particularly in the model resulting from the combination of acoustic and linguistic features for the story retelling task, where only two linguistic features (exclamation and conjunction) were included. This behavior is also clearly visible in Table 6, where the results of models trained only on linguistic features were significantly worse compared to those trained exclusively on acoustic features and never exceeded the values of models trained on acoustic features. The most influential acoustic features across Figure 8, Figure 9 and Figure 10 were features related to MFCC, audSpec_Rfilt, pcm_fftMag, and shimmer. Interestingly, the models shown in Figure 8 and Figure 9 also utilized information about the presence of silence in the speech of the given person.

Subsequently, we investigated Shapley values of the top-performing XGBoost models trained on the whole dataset employing identified important features, and we present the results in Figure 11, Figure 12 and Figure 13. For XGBoost models applied in story retelling tasks, there are two features that retain their high impact on the prediction as top-three important features:audSpec_Rfilt_sma[5]_mean for the model trained on acoustic and linguistic features, and shimmerLocal_sma_std for the model trained on acoustic features. The feature silences_number_ratio becomes more important for both XGBoost models in story retelling tasks, confirming that longer silence segments during speech are attributed to PD patients. Similarly, pcm_fftMag_spectralRollOff25.0_sma_mean advances as the third most important feature according to SHAP values, with lower feature values suggesting PD patients. For the model trained on acoustic and linguistic features, two important linguistic features (CCONJ_POS, DET_POS) gain slightly more relevance and are located among the top five features with the highest mean absolute SHAP values. For the XGBoost model applied to the word “pataka” in the diadochokinetic task, shimmerLocal_sma_std has the highest importance according to SHAP values, similar to the model trained on acoustic features in the story retelling task. The following most important features are audSpec_Rfilt_sma[0]_median and pcm_fftMag_spectralKurtosis_sma_mean.

Finally, for the purpose of final feature visualizations, box plots were created showing the 10 most important features (repeating in 5-fold cross-validation). These plots depict the differences between the group of individuals with Parkinson’s disease and the control group to assess how significantly the two groups differ. These plots are presented in Figure 14, Figure 15 and Figure 16 for models that achieved the best results for specific metrics such as recall, precision, and F1-score. Again, these were models for:Acoustic and linguistic features for the task: story retellingAcoustic features for the task: story retellingThe diadochokinetic task for the word “pataka”.

What can be observed in Figure 14, Figure 15 and Figure 16 is the high variability of features between both groups, which is based on features such as silences_number_ratio and specific MFCC and audSpec parameters. Additionally, the control group (HC) shows higher median values and greater variability for many features, which may indicate greater diversity in normal speech patterns compared to the more uniform speech characteristics observed in patients with Parkinson’s disease, confirming the hypothesis of the monotony of speech in individuals with Parkinson’s disease.

### 3.4. Deep Learning Classification Results

The results obtain from language models were shown in Table 7. The metrics given are average results with standard deviations for the 5 folds on which the study was conducted. The highest metrics are presented in bold font. The best results were achieved for diadochokinetic task. Compared to the rest of the audio for all models the metrics are often more than 10% higher for “pakata”, “pekata”, and “petaka” recordings. The highest F1-score was achieved by the HuBERT base model, which was 0.94 with a deviation of 0.04. HuBERT and wav2vec performed worst for the D1 task (picture description), with the wav2vec-base model showing the worst performance with 0.62 ± 0.14 F1-score. WavLM got the worst result for all recordings which was 0.69 ± 0.08. Considering the F1-score value, most of the best results for different types of tasks were achieved by the HuBERT-base model. The only exception is the AQ task (daily activity description), where wav2vec2-base achieved a result 1 percent higher and a three times smaller standard deviation. The second-best-performing model is wav2vec-base, which outperformed WavLM-base in all tasks except DI. Although the results for diacokinesis are the same, the standard deviation is lower.

## 4. Discussion

This study developed a Parkinson’s disease (PD) detection algorithm using speech data, emphasizing the importance of both acoustic and linguistic features in reliably distinguishing PD patients from healthy controls (HC). The process began with speech signal preprocessing (including echo cancellation), followed by feature extraction, feature selection, and classification. Notably, data were collected with a Microsoft HoloLens 2 head-mounted display (HMD) during diverse speech tasks such as picture description, story retelling, and vowel phonation. This setup offered a consistent microphone placement and demonstrated the feasibility of integrating mixed reality (MR) technology into clinical workflows. Key extracted features—such as formants, silence intervals, and spectral parameters like Mel-frequency cepstral coefficients (MFCCs) and shimmer—underwent thorough examination, allowing us to capture subtle but telling clues about patients’ vocal timbre, prosody, and voice stability.

From the acoustic perspective, analysis revealed that speech tasks involving spontaneous monologues (e.g., story retelling) provided the clearest differentiation between PD and HC groups. In these tasks, MFCCs, PCM FFT magnitudes, and shimmer emerged as major contributors to detecting typical PD-associated traits such as reduced pitch variation, breathiness, and vocal fatigue. The ratio and duration of silence segments also stood out as influential markers, especially for identifying slowed articulation or festinating speech patterns often reported in PD. Meanwhile, linguistic features—like part-of-speech usage or the presence of function words—displayed some predictive power but appeared less decisive than acoustic cues. The collective evidence underlines that multifaceted vocal biomarkers provide a richer representation of neurological status, capturing both how patients speak (acoustics) and what they say (linguistics).

Regarding our modeling approaches, we employed traditional machine learning (ML) classifiers (e.g., SVM, random forests) and state-of-the-art deep learning (DL) architectures (e.g., wav2vec2, HuBERT). Among ML models, XGBoost consistently outperformed others, often surpassing or matching published state-of-the-art (SOTA) results, as summarized in Table 8. The story-retelling task that integrated acoustic and linguistic features achieved an impressive mean F1-score of 0.90, accompanied by high recall and precision, and stable performance across cross-validation folds. Such robust metrics underscore not only the viability of these markers for PD detection but also the algorithm’s capacity to generalize well to new cases. The diadochokinetic task also proved effective, particularly when analyzed using the deep learning frameworks; for instance, HuBERT and wav2vec2 attained F1-scores that rivaled or even surpassed some benchmarks in the literature. This exceptional performance illustrates how DL models can capture the nuanced, high-dimensional patterns associated with hypokinetic dysarthria in PD.

These findings hold direct relevance for clinical practice, where early detection and continuous monitoring of PD remain critical objectives. In typical workflows, disease severity is rated using clinical scales like the MDS-UPDRS, which, while vital, can be subjective and is usually performed at sparse intervals. By contrast, the proposed approach provides *objective, quantifiable* speech metrics that can be recorded frequently, even remotely. This real-time or at-home voice collection could enable more sensitive, timely detection of disease onset—particularly in individuals who exhibit subtle speech deviations well before they develop pronounced motor symptoms. Additionally, clinicians can use these voice metrics in tandem with MDS-UPDRS scores; for instance, a stable MDS-UPDRS motor score paired with worsening vocal biomarkers might signal an unrecognized shift in the patient’s condition, prompting earlier intervention or more detailed evaluations.

Furthermore, because speech tasks can be easily administered in a clinic or remotely through smartphones and HMDs like HoloLens 2, patients could perform short voice tests daily or weekly without cumbersome appointments. Such scalable, non-invasive screening offers a practical method for telemedicine, enabling specialists to monitor progression and medication efficacy from afar. The combination of acoustic and linguistic analyses also promotes interpretability—measurements like decreased pitch variation or increased silence durations align with standard clinical observations of PD-related dysarthria. This interpretable synergy between computational outputs and clinicians’ expertise fosters trust in the system and accelerates its integration into routine care. Ultimately, these methods can help personalize treatment strategies: therapy decisions, dosage adjustments, or targeted speech therapy can be based on real-time insights into each patient’s specific vocal profile.

To ensure widespread clinical adoption, future efforts could expand patient cohorts and incorporate different PD phenotypes, medication regimens, and disease stages. Larger longitudinal studies can validate the sensitivity of these vocal biomarkers for detecting minimal or prodromal PD symptoms, as well as tracking advanced-stage patients. Moreover, combining speech metrics with other sensors (e.g., hand, head, and gait motion from MR devices) may yield even more robust multi-modal markers, enhancing diagnostic power. In parallel, developers should improve the explainability of DL solutions, ensuring clinicians understand how specific acoustic changes link to PD physiology. By continuing these refinements, the framework presented here stands to become a valuable, patient-friendly asset in the clinical management of Parkinson’s disease—advancing both early detection and personalized, data-driven care.

**Table 8 sensors-25-02405-t008:** Comparison of Parkinson’s disease detection methods.

Study	Dataset Size	Method (ML/DL)	Task	Best F1-Score, Recall, Precision
Our Work	57 (21 PD, 36 HC)	ML: Logistic Regression, SVM, Random Forests, AdaBoost, XGBoost	Story Retelling (Acoustic + Linguistic)	**F1:** 0.90 ± 0.05 **Recall:** 0.91 ± 0.11 **Precision:** 0.92 ± 0.10
Our Work	57 (21 PD, 36 HC)	DL: HuBERT, WavLM, wav2vec 2.0	Diadochokinetic Task	**F1:** 0.94 ± 0.04 **Recall:** 0.92 ± 0.08 **Precision:** 0.98 ± 0.03
[15]	223 (115 PD, 108 HC)	ML: SVM	Standardized Italian Sentence	**F1:** 0.79 **Recall:** 0.83 **Precision:** 0.75
[17]	338 (72 Early PD, 266 HC)	ML: KNN, SVM, NB, DL: CNN	Articulation of the Vowel /e/	**F1:** 0.82 ± 0.01 **Recall:** 0.83 ± 0.01 **Precision:** 0.81 ± 0.01
[17]	354 (88 Mid-Advanced PD, 266 HC)	ML: KNN, SVM, NB, DL: CNN	Articulation of the Vowel /e/	**F1:** 0.80 ± 0.03 **Recall:** 0.80 ± 0.01 **Precision:** 0.79 ± 0.03
[20]	37 (16 PD, 21 HC)	ML: Cubic SVM, Fine KNN, DL: Wide neural network	Reading Task	**F1:** 0.92 **Recall:** 0.91 **Precision:** 0.93
[20]	37 (16 PD, 21 HC)	ML: Cubic SVM, Fine KNN, DL: Wide neural network	Spontaneous Conversation	**F1:** 0.90 **Recall:** 0.90 **Precision:** 0.90
[23]	65 (28 PD, 37 HC)	DL: YAMNet	Standardized Italian Sentence	**F1:** 0.84 **Recall:** 0.74 **Precision:** 1.00
[21]	188 (124 PD, 64 HC)	DL: CNN-LSTM	Articulation of the Vowel /a/	**F1:** 0.96 **Recall:** 0.96 **Precision:** 0.95
[22]	252 (188 PD, 64 HC)	DL: LSTM	Articulation of the Vowel /a/	**F1:** 0.93 **Recall:** 0.90 **Precision:** 0.96

## 5. Conclusions

This study has demonstrated a novel data acquisition protocol for speech and voice tasks in Parkinson’s disease (PD) using mixed reality (MR) technology. Compared to conventional recording setups, the MR head-mounted display (HMD) offers a consistent microphone distance and facilitates the potential integration of additional data modalities such as hand motion, gaze tracking, and full-body movement. Our results confirm the feasibility and utility of MR-based voice collection for high-quality acoustic and linguistic analysis—key steps toward multimodal neurodegenerative disease diagnostics.

A specialized medical processing pipeline was developed to identify and quantify vocal biomarkers across various tasks, including monologue-type descriptions, vowel phonations, and diadochokinetic utterances. In machine learning experiments, the XGBoost model attained particularly high performance (F1 = 0.90) in the story-retelling task, highlighting the importance of acoustic parameters such as MFCC-related metrics, spectral features, and shimmer in differentiating PD patients from healthy controls. Furthermore, deep learning methods—particularly wav2vec2, HuBERT, and WavLM—showed substantial diagnostic power, matching or surpassing select state-of-the-art benchmarks for detecting PD from spoken data.

Beyond confirming the relevance of acoustic and linguistic biomarkers in PD, this work underscores the synergy between advanced speech-processing models and stable, well-controlled MR data acquisition. This combined approach can accelerate the development of accessible, reliable, and explainable PD screening tools. Future extensions may incorporate motor and visual measures captured by MR devices, ultimately leading to a truly multimodal framework for early detection and monitoring of Parkinson’s disease. By integrating speech biomarkers with additional sensor data, clinicians can gain deeper insight into disease progression, refine treatment strategies, and potentially enhance patient outcomes through more frequent and objective assessments.

## Figures and Tables

**Figure 1 sensors-25-02405-f001:**
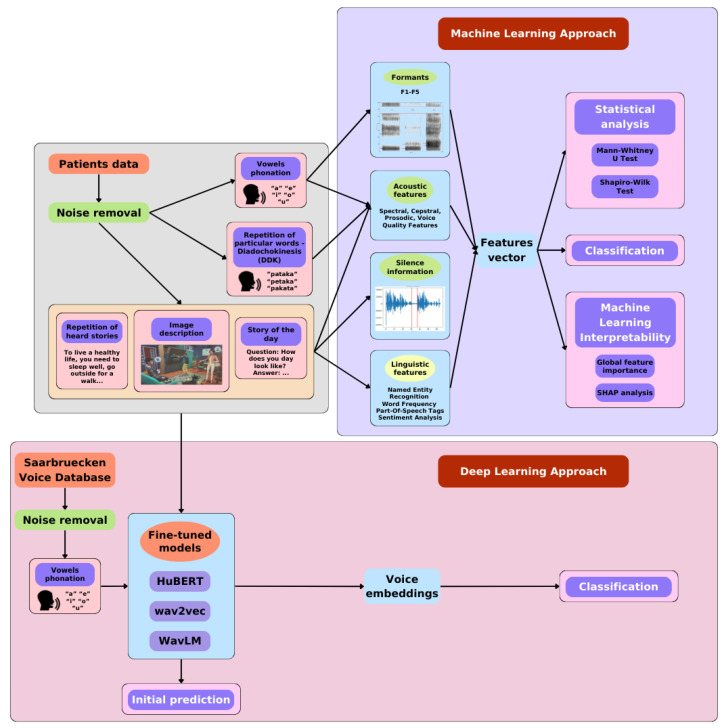
The workflow in machine and deep learning methods.

**Figure 2 sensors-25-02405-f002:**
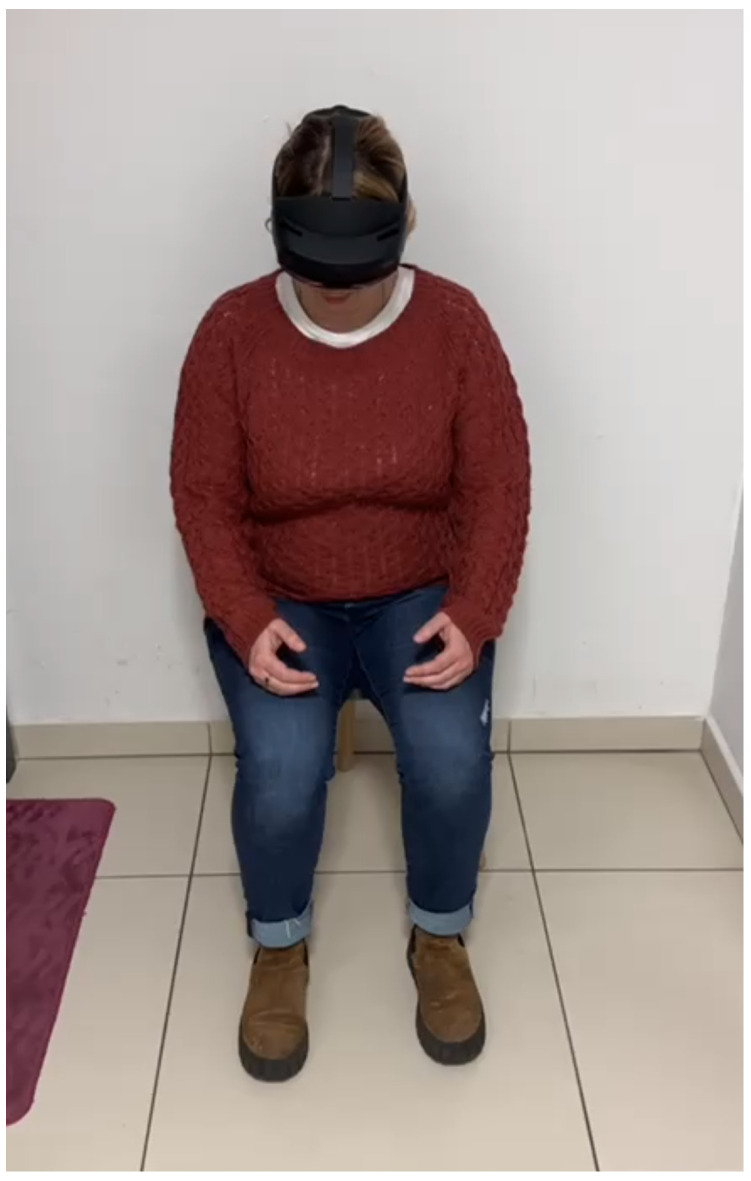
Patient testing the MS HoloLens 2 MR HMD.

**Figure 3 sensors-25-02405-f003:**
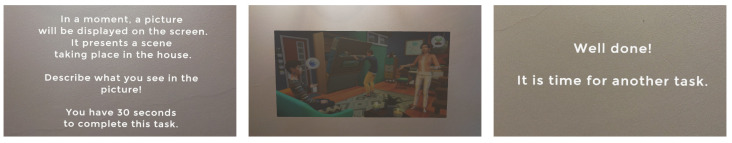
Voice task view from a user perspective: exemplary task instruction, presented in text and read by a lector; presented content of a task; MR screen display showing task completion.

**Figure 4 sensors-25-02405-f004:**
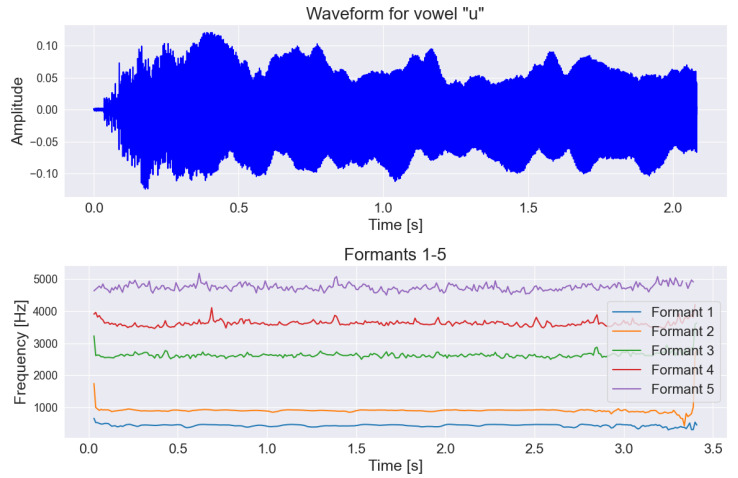
Graph of the first formants shown on the frequency-time chart along with the original voice signal for a person with Parkinson’s disease during the pronunciation of the vowel “u”.

**Figure 5 sensors-25-02405-f005:**
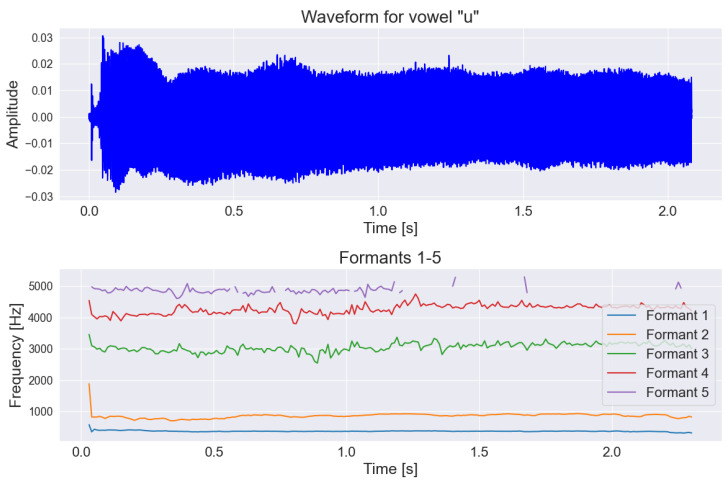
Graph of the first formants shown on the frequency-time chart along with the original voice signal for a person from control group during the pronunciation of the vowel “u”.

**Figure 6 sensors-25-02405-f006:**
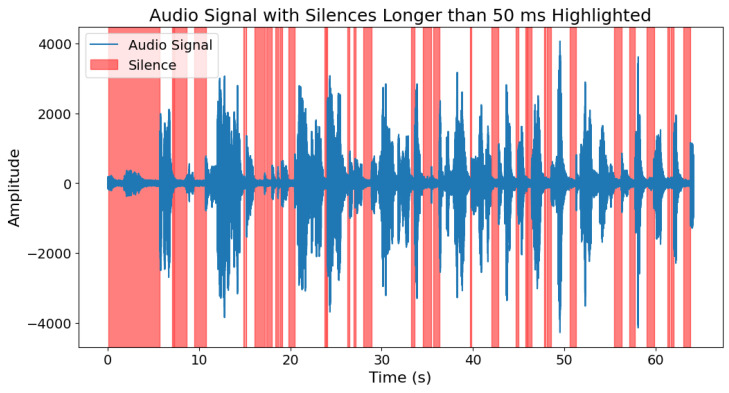
Silence longer than 50 ms within the speech of a person with Parkinson’s disease.

**Figure 7 sensors-25-02405-f007:**
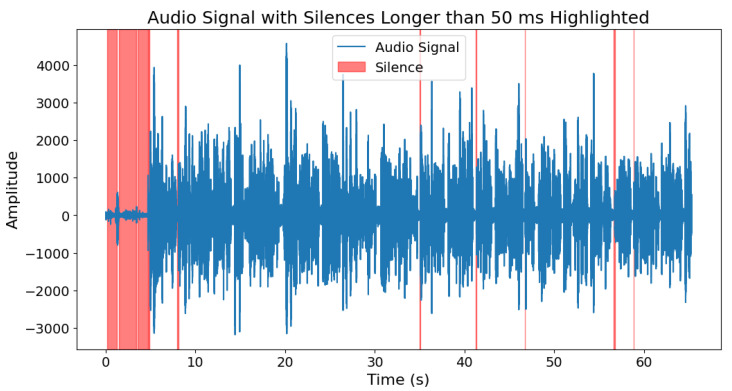
Silence longer than 50 ms within the speech of a person from control group.

**Figure 8 sensors-25-02405-f008:**
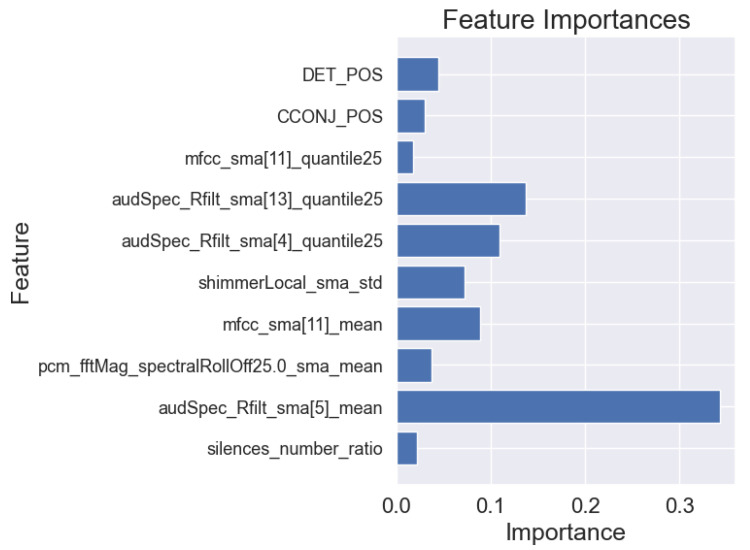
Importance of features of the XGBoost model for 5-fold cross-validation for acoustic and linguistic features in the task: story retelling.

**Figure 9 sensors-25-02405-f009:**
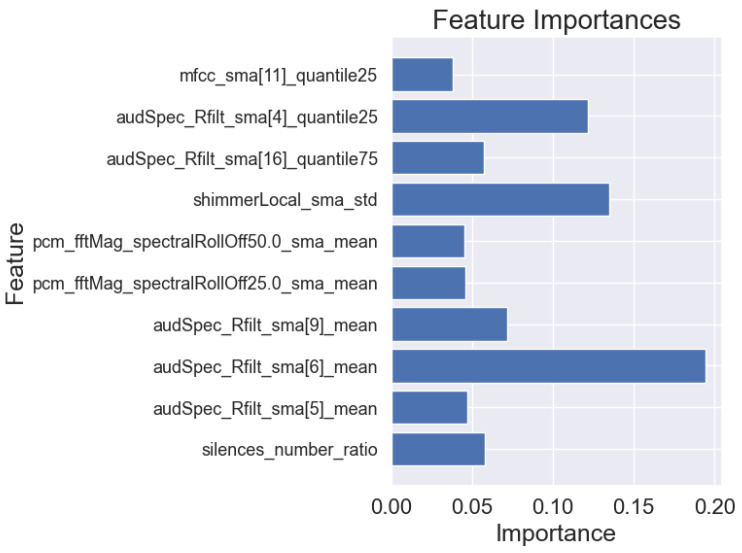
Importance of features of the XGBoost model for 5-fold cross-validation for acoustic features in the task: story retelling.

**Figure 10 sensors-25-02405-f010:**
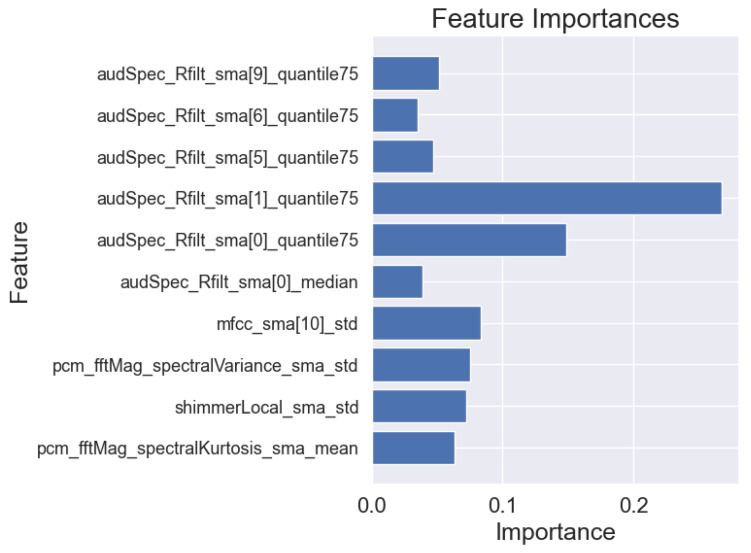
Importance of features of the XGBoost model for 5-fold cross-validation for the word “pataka” in the diadochokinetic task.

**Figure 11 sensors-25-02405-f011:**
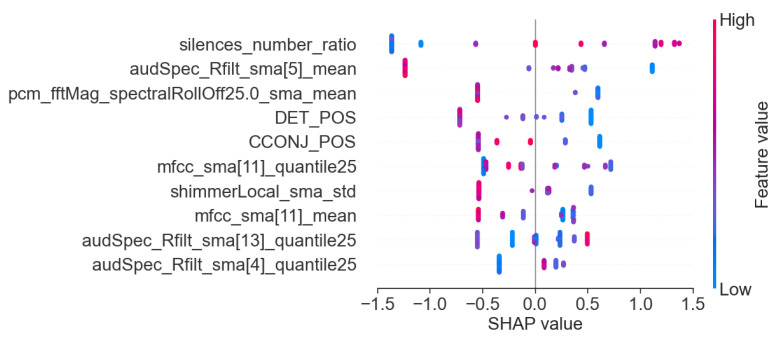
SHAP values of the XGBoost model trained on important acoustic and linguistic features in the task: story retelling.

**Figure 12 sensors-25-02405-f012:**
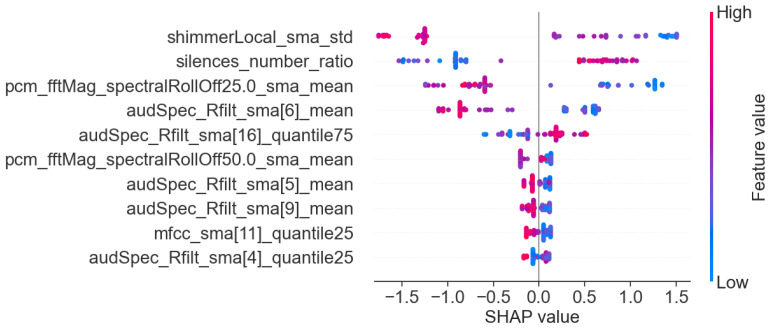
SHAP values of the XGBoost model trained on important acoustic features in the task: story retelling.

**Figure 13 sensors-25-02405-f013:**
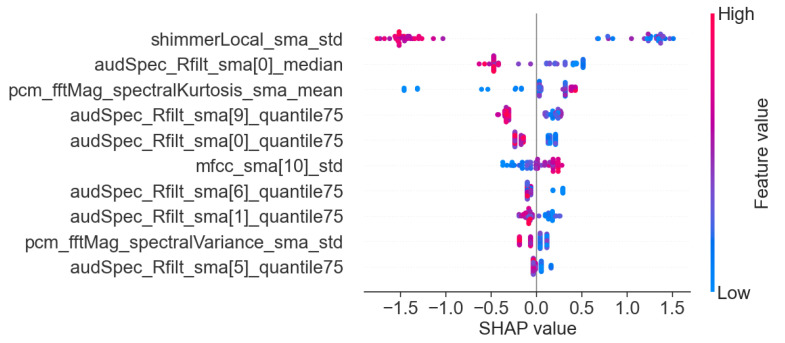
SHAP values of the XGBoost model trained on important features for the word ”pataka” in the diadochokinetic task.

**Figure 14 sensors-25-02405-f014:**
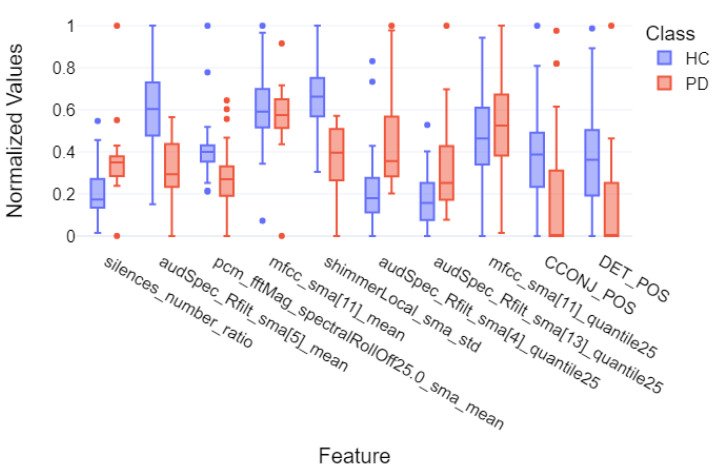
Box plot of the 10 most important features of the XGBoost model for 5-fold cross-validation divided into individual groups for the task of story retelling—for the linguistic and acoustic parts.

**Figure 15 sensors-25-02405-f015:**
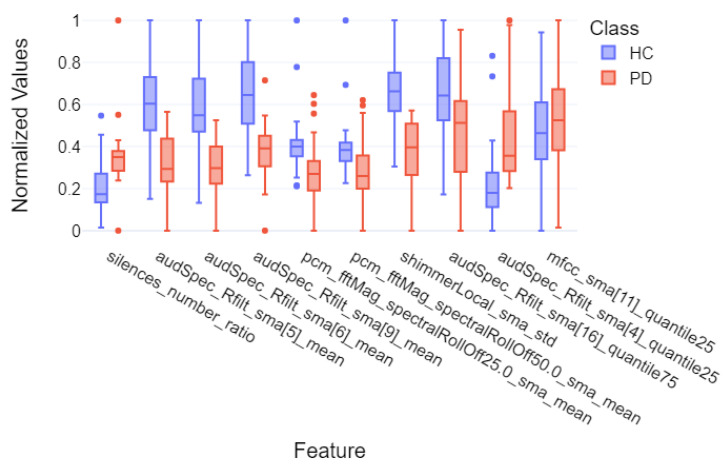
Box plot of the 10 most important features of the XGBoost model for 5-fold cross-validation divided into individual groups for the task of story retelling—acoustic part.

**Figure 16 sensors-25-02405-f016:**
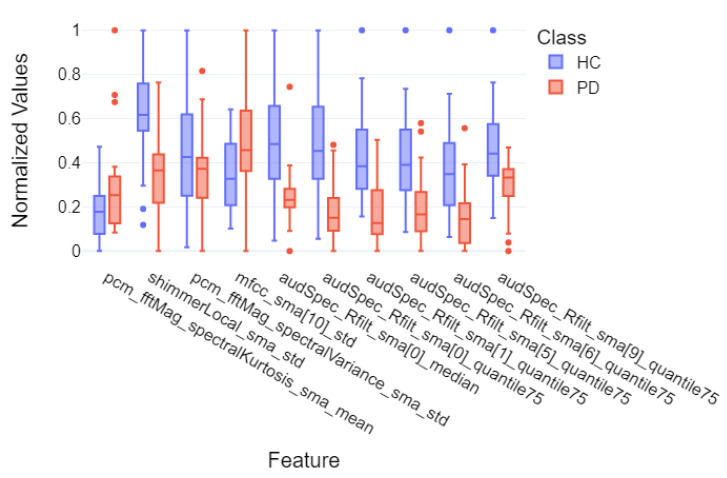
Box plot of the 10 most important features of the XGBoost model for 5-fold cross-validation divided into individual groups for the diadochokinetic task—word “pataka”.

**Table 1 sensors-25-02405-t001:** Low-level descriptors (LLDs) and their grouping.

Low-Level Descriptors	Group
**Energy**	
Sum of auditory spectrum	Prosodic
Sum of RASTA-filtered auditory spectrum	Prosodic
RMS Energy	Prosodic
Zero Crossing Rate	Prosodic
**F0 (Pitch)**	
Fundamental frequency	Prosodic
Probability of Voicing	Voice Quality
**Voicing Quality**	
Jitter (Local)	Voice Quality
Jitter (Delta)	Voice Quality
Shimmer (Local)	Voice Quality
**RASTA and Spectral Features**	
RASTA-style filtered auditory spectral bands [1–26]	Spectral
Spectral Flux	Spectral
Spectral Entropy	Spectral
Spectral Variance	Spectral
Spectral Skewness	Spectral
Spectral Kurtosis	Spectral
Spectral Slope	Spectral
Spectral Roll-Off (0.25, 0.50, 0.75, 0.9)	Spectral
Spectral Energy 25–650 Hz	Spectral
Spectral Energy 1–4 kHz	Spectral
**MFCC**	
Mel-Frequency Cepstrum Coefficients	Cepstral

**Table 2 sensors-25-02405-t002:** Number of significant features out of all features derived for each task and number of significant features recurrent in each task within specified task groups. DI—picture description, AQ—daily activity description, RS—story retelling, V—vowel phonation, DDK—diadochokinetic task, NLP—linguistic features, AC—acoustic features.

Task		No. of Significant Features
Monologue AC	DI	122/399
	AQ	182/399
	RS	257/399
Recurrent		113
Monologue NLP	DI	3/31
	AQ	8/30
	RS	8/30
Recurrent		1
V	A	73/425
	E	64/425
	I	68/425
	O	85/425
	U	40/425
Recurrent		9
DDK	pataka	159/390
	petaka	172/390
	pakata	219/390
Recurrent		134

**Table 3 sensors-25-02405-t003:** Mann–Whitney U test results and effect size measured with Cohen’s *d* for important features of the XGBoost model for 5-fold cross-validation for the word “pataka” in the diadochokinetic task.

Feature	Test Statistic *U*	*p* Value	Cohen’s *d*	Effect Size
pcm_fftMag_spectralKurtosis_sma_mean	262.0	0.1016	−0.54	medium
shimmerLocal_sma_std	630.0	<0.0001	1.58	very large
pcm_fftMag_spectralVariance_sma_std	412.0	0.3451	0.29	small
mfcc_sma[10]_std	220.0	0.0180	−0.76	medium
audSpec_Rfilt_sma[0]_median	595.0	<0.0001	1.26	very large
audSpec_Rfilt_sma[0]_quantile75	624.0	<0.0001	1.52	very large
audSpec_Rfilt_sma[1]_quantile75	606.0	<0.0001	1.36	very large
audSpec_Rfilt_sma[5]_quantile75	573.0	0.0002	1.13	large
audSpec_Rfilt_sma[6]_quantile75	586.0	0.0001	1.17	large
audSpec_Rfilt_sma[9]_quantile75	569.0	0.0003	1.13	large

**Table 4 sensors-25-02405-t004:** Mann–Whitney U test results and effect size measured with Cohen’s *d* for important features of the XGBoost model for 5-fold cross-validation for acoustic and linguistic features in the task: story retelling.

Feature	Test Statistic *U*	*p* Value	Cohen’s *d*	Effect Size
silences_number_ratio	135.0	0.0001	−1.08	large
audSpec_Rfilt_sma[5]_mean	672.0	<0.0001	1.64	very large
pcm_fftMag_spectralRollOff25.0_sma_mean	572.0	0.0014	0.79	medium
mfcc_sma[11]_mean	416.0	0.5350	0.24	small
shimmerLocal_sma_std	699.0	<0.0001	1.91	very large
audSpec_Rfilt_sma[4]_quantile25	118.0	<0.0001	−1.18	large
audSpec_Rfilt_sma[13]_quantile25	205.0	0.0043	−0.98	large
mfcc_sma[11]_quantile25	305.0	0.2304	−0.28	small
CCONJ_POS	558.5	0.0026	0.72	medium
DET_POS	583.0	0.0006	0.89	large

**Table 5 sensors-25-02405-t005:** Mann–Whitney U test results and effect size measured with Cohen’s *d* for important features of the XGBoost model for 5-fold cross-validation for acoustic features in the task: story retelling.

Feature	Test Statistic *U*	*p* Value	Cohen’s *d*	Effect Size
silences_number_ratio	135.0	0.0001	−1.07	large
audSpec_Rfilt_sma[5]_mean	672.0	<0.0001	1.64	very large
audSpec_Rfilt_sma[6]_mean	670.0	<0.0001	1.59	very large
audSpec_Rfilt_sma[9]_mean	651.0	<0.0001	1.52	very large
pcm_fftMag_spectralRollOff25.0_sma_mean	572.0	0.0014	0.79	medium
pcm_fftMag_spectralRollOff50.0_sma_mean	558.0	0.0030	0.67	medium
shimmerLocal_sma_std	699.0	<0.0001	1.91	very large
audSpec_Rfilt_sma_quantile75	540.0	0.0075	0.84	large
audSpec_Rfilt_sma[4]_quantile25	118.0	<0.0001	−1.18	large
mfcc_sma[11]_quantile25	305.0	0.2304	−0.28	small

**Table 6 sensors-25-02405-t006:** Classification results of XGBoost classifiers trained on different sets of features in all tasks. 5F—5-fold F1-score, rec—recall, prec—precision, LOO—leave-one-out, DI—picture description, AQ—daily activity description, RS—story retelling, V—vowel phonation, DDK—diadochokinetic task, NLP—linguistic features, AC—acoustic features.

Task	5F Recall	5F Precision	5F F1-Score
DI AC	0.93 ± 0.13	0.88 ± 0.15	0.89 ± 0.09
DI NLP	0.65 ± 0.08	0.81 ± 0.24	0.71 ± 0.14
DI all	0.93 ± 0.13	0.88 ± 0.15	0.89 ± 0.09
AQ AC	0.90 ± 0.12	0.84 ± 0.13	0.85 ± 0.03
AQ NLP	0.47 ± 0.07	0.68 ± 0.28	0.54 ± 0.13
AQ all	0.90 ± 0.12	0.79 ± 0.11	0.83 ± 0.05
**RS AC**	**0.95 ± 0.10**	0.86 ± 0.13	0.89 ± 0.07
RS NLP	0.67 ± 0.10	0.95 ± 0.10	0.78 ± 0.07
**RS all**	0.91 ± 0.11	0.92 ± 0.10	**0.90 ± 0.05**
V A	0.75 ± 0.22	0.89 ± 0.14	0.80 ± 0.16
V E	0.70 ± 0.19	0.90 ± 0.12	0.77 ± 0.12
V I	0.76 ± 0.10	0.85 ± 0.12	0.77 ± 0.06
V O	0.80 ± 0.19	0.88 ± 0.16	0.82 ± 0.13
V U	0.55 ± 0.19	0.92 ± 0.16	0.65 ± 0.15
V all	0.80 ± 0.19	0.91 ± 0.14	0.83 ± 0.14
**DDK pataka**	0.83 ± 0.24	**0.96 ± 0.08**	0.86 ± 0.16
DDK petaka	0.73 ± 0.25	0.96 ± 0.08	0.80 ± 0.16
DDK pakata	0.77 ± 0.31	0.84 ± 0.13	0.76 ± 0.23
DDK all	0.88 ± 0.24	0.83 ± 0.11	0.85 ± 0.17
Overall	0.80 ± 0.27	0.95 ± 0.10	0.83 ± 0.18

**Table 7 sensors-25-02405-t007:** Classification of MR-based voice and speech data for Parkinson’s disease using DNN. The description of the tasks is the same as in Table 6.

HuBERT-Base Results			
**Task**	**5F Recall**	**5F Precision**	**5F F1-Score**
DI	0.71 ± 0.06	0.73 ± 0.12	0.72 ± 0.11
AQ	0.85 ± 0.07	0.80 ± 0.12	0.80 ± 0.03
RS	0.85 ± 0.07	0.80 ± 0.12	0.80 ± 0.03
V	0.82 ± 0.08	0.80 ± 0.14	0.79 ± 0.06
**DDK**	**0.92 ± 0.08**	**0.98 ± 0.03**	**0.94 ± 0.04**
all	0.76 ± 0.21	0.86 ± 0.19	0.75 ± 0.12
**WavLM-Base Results**			
**Task**	**5F Recall**	**5F Precision**	**5F F1-Score**
DI	0.80 ± 0.14	0.68 ± 0.04	0.72 ± 0.08
AQ	0.86 ± 0.03	0.71 ± 0.08	0.75 ± 0.05
RS	0.88 ± 0.07	0.72 ± 0.09	0.78 ± 0.04
V	0.76 ± 0.07	0.71 ± 0.05	0.72 ± 0.03
**DDK**	**0.89 ± 0.07**	**0.93 ± 0.09**	**0.90 ± 0.06**
all	0.71 ± 0.15	0.88 ± 0.19	0.69 ± 0.08
**wav2vec2-Base Results**			
**Task**	**5F Recall**	**5F Precision**	**5F F1-Score**
DI	0.60 ± 0.20	0.73 ± 0.13	0.62 ± 0.14
AQ	0.87 ± 0.04	0.77 ± 0.03	0.81 ± 0.01
RS	0.84 ± 0.02	0.79 ± 0.01	0.80 ± 0.01
**V**	**0.96 ± 0.09**	0.64 ± 0.20	0.73 ± 0.16
**DDK**	0.90 ± 0.11	**0.94 ± 0.08**	**0.90 ± 0.11**
all	0.65 ± 0.07	0.86 ± 0.20	0.67 ± 0.07

## Data Availability

Due to ethical restrictions, our data collected from Microsoft Hololens 2 head-mounted display will be shared on request.

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
