# Peer review of "Analysis of Voice, Speech, and Language Biomarkers of Parkinson’s Disease Collected in a Mixed Reality Setting"

_sensors, 2025, doi:10.3390/s25082405_

Round 1

Reviewer 1 Report

Comments and Suggestions for Authors

See attached file

Author Response

Dear Reviewer,

Thank you for your thoughtful comments on our manuscript. We fully agree with your observation that voice analysis alone is not sufficient for diagnosing neurodegenerative diseases. Rather, our aim is to highlight how this non-invasive, easily accessible screening tool can complement more comprehensive diagnostic methods. Below, we address your specific points in detail and explain how we have either integrated them into the revised manuscript or will consider them in future research.

Comments 1:

Actually, I believe that a diagnosis of neurodegenerative disease will never be based on vocal analysis. This may be a “support” to diagnostics, a tool to be added to refine the diagnosis, or at least a method for early screening of neurodegeneration.”

Response 1: We completely concur that neurodegenerative disease diagnosis must rely on a range of clinical and diagnostic inputs to achieve the highest accuracy. We envision voice analysis primarily as a first-line screening tool or an adjunct to clinical assessments. By flagging subtle changes early, voice analysis can prompt timely specialist evaluation. This approach is particularly important in resource-limited settings or in situations where regular, large-scale screening may be beneficial. 

Comments 2:

Another possible application is on the follow-up of patients, in order to reveal signs of disease impairment in the home environment. However, in this context the use of sophisticated devices may not be practical, and we should expect significant performance impairment due to suboptimal devices and environmental conditions.

Response 2: We acknowledge that the use of sophisticated devices at home introduces potential challenges, including background noise, environmental interference, and varied recording equipment. To address these issues, our current research involves collecting hundreds of hours of speech data in realistic conditions to enhance algorithmic resilience. We draw parallels with commercially successful automatic speech recognition (ASR) systems that have achieved robust performance in similar circumstances. By leveraging large-scale data and advanced signal processing techniques, we aim to ensure that our method remains reliable in diverse real-world environments. 

Comments 3:

Not to exclusively focus on diagnosis but at least discuss the potential of their algorithms to perform a multi-class classification that is related e.g. to the MDS-UPDRS level of patients.

Response 3: We appreciate your suggestion to extend our detection algorithm to a multi-class classification framework, especially to capture distinct disease stages aligned with MDS-UPDRS levels. Our current study focuses on establishing proof of concept for robust Parkinson’s disease (PD) detection through voice features due to limited number of patients. However, we recognize that multi-class classification could provide a more nuanced understanding of disease progression. In future work, we intend to investigate how both acoustic and linguistic markers evolve across mild, moderate, and advanced PD, enabling us to map changes in speech parameters more precisely to clinical severity.

Comments 4:

To better describe the clinical conditions of the included patients. I understand that average/std values are reported, but more detailed information (e.g., how many patients are classified as moderate/advanced in terms of HY or UPDRS) could be useful to better appreciate the potential of the technique.

Response 4: Thank you for highlighting the importance of presenting a detailed clinical profile of our patient cohort. We have revised the manuscript (Materials and Methods/Patient data, lines: 230-236; 246-256; 259-273) to include specific information regarding the disease stages of our participants (e.g., mild, moderate, and advanced cases). This additional detail will provide clearer context for interpreting our algorithm’s performance and understanding its potential applicability across different PD stages. We believe this amendment will offer more transparency and assist readers in evaluating the broader clinical implications of our findings.

Comments 5:

Finally, please mention the Ethical Approval number, as you are working with patients.

Response 5: Thank you for highlighting this important point. We confirm that the ethics approval number is included in the Institutional Review Board Statement section in our manuscript. This ensures transparency and compliance with ethical standards in patient-related research.

Thank you again for your constructive feedback. We share your perspective on the critical need for a multi-modal diagnostic approach and the importance of robust data collection in real-world settings. Your insights have helped strengthen our manuscript, and we look forward to refining our methods further to ensure the practical applicability of voice-based detection and monitoring of neurodegenerative diseases.

Kind regards,
Milosz Dudek

AGH University of Krakow

Reviewer 2 Report

Comments and Suggestions for Authors

Dear Authors,

I have had the pleasure of reviewing your work from my perspective as a clinical neurologist specializing in movement disorders. First, I would like to congratulate you on your well-executed study and innovative approach to detecting symptoms of Parkinson's disease. Integrating mixed reality technology with speech analysis represents a novel contribution to the field. Standardizing the setup addresses a significant methodological challenge in speech analysis studies and ensures the accuracy of your metrics. Your comprehensive analysis of machine learning and deep learning techniques has yielded impressive results, with the XGBoost model's F1-score at 0.90 and the Hubert model's F1-score at 0.94. I see real potential for clinical applications, especially considering the non-invasive nature of speech assessment.

As a clinician who regularly sees Parkinson's patients, I am particularly intrigued by your identification of specific acoustic features as significant markers for differentiating PD patients. The silence intervals during speech align with clinical observations of PD patients, who often exhibit hesitations and speech festination.

From a clinical perspective, I believe your method has the potential for both early detection and monitoring of Parkinson's disease. The ability to objectively quantify speech changes could complement our somewhat subjective clinical scales and potentially detect subtle changes before pronounced motor symptoms develop.

However, to strengthen the clinical applicability of your work, I encourage you to explore several additional aspects:

1. The correlation between your speech biomarkers and established clinical metrics is valuable. Although you collected MDS-UPDRS-III scores, VHI, and PL-VDCQ data, analyzing how these correlate with your speech parameters will validate their clinical relevance.

2.  As a clinician, I am curious how these speech patterns may vary across disease stages and phenotypes. Stratifying patients according to disease severity or dominant symptom profile (tremor-dominant versus akinetic-rigid) could offer insights that enhance clinical interpretation.

3. Medication status significantly influences all aspects of PD symptomatology in clinical practice. Understanding whether patients were evaluated in ON or OFF states and how levodopa may affect these speech parameters would improve clinical applicability.

4. Conducting longitudinal assessments would be especially valuable in determining how these biomarkers correlate with disease progression, which is crucial for monitoring patients over time.

Thank you for your significant contributions to this important area of research. I look forward to seeing how your work evolves.

I am excited to see how you further develop this promising line of research.

Kind regards.

Author Response

Dear Reviewer,

We sincerely appreciate your detailed and thoughtful review of our manuscript. Your expertise as a clinical neurologist specializing in movement disorders has provided invaluable insights into the clinical relevance of our work. We are encouraged by your positive feedback regarding our novel integration of mixed reality technology with speech analysis and the promising results achieved by our machine learning models. Below, we address each of your specific suggestions and describe the corresponding revisions or planned future work.

Comments 1:

The correlation between your speech biomarkers and established clinical metrics is valuable. Although you collected MDS-UPDRS-III scores, VHI, and PL-VDCQ data, analyzing how these correlate with your speech parameters will validate their clinical relevance.

Response 1: Thank you for underscoring the importance of correlating speech biomarkers with established clinical metrics. To validate the clinical significance of our findings, we examined correlations between the acoustic features and the MDS-UPDRS-III and MoCA scales. Our preliminary analysis yielded the following most correlated parameters:

  • Most correlated with MoCA

    • pakata_F0final_sma_quantile50: 0.736897

    • pakata_F0final_sma_median: 0.736897

    • u_F0final_sma_quantile50: 0.628014

    • u_F0final_sma_median: 0.628014

    • u_F0final_sma_quantile25: 0.611922

    • pataka_F0final_sma_std: 0.576039

    • o_F0final_sma_quantile25: 0.569187

    • pakata_F0final_sma_mean: 0.528496

    • petaka_F0final_sma_std: 0.525408

  • Most correlated with MDS-UPDRS-III

    • o_F0final_sma_std: 0.623758

    • e_F0final_sma_std: 0.406747

    • u_F0final_sma_std: 0.396065

    • e_F0final_sma_quantile75: 0.293369

    • a_F0final_sma_std: 0.280647

    • u_F0final_sma_quantile75: 0.161302

    • Describe-story_F0final_sma_quantile75: 0.122610

    • e_F0final_sma_mean: 0.122224

These results have been incorporated into the revised manuscript (Results/Statistical analysis section, lines: 477-521) to highlight the clinical relevance of our speech biomarkers, and we further discuss their potential implications in the Discussion section.

Comments 2:

As a clinician, I am curious how these speech patterns may vary across disease stages and phenotypes. Stratifying patients according to disease severity or dominant symptom profile (tremor-dominant versus akinetic-rigid) could offer insights that enhance clinical interpretation.

Response 2: We appreciate your insightful recommendation to investigate speech variation across different Parkinson’s disease (PD) phenotypes and disease stages. We acknowledge that our current sample size limits the ability to perform a robust stratification. However, we have updated the Discussion to emphasize the importance of this analysis and to outline our plans for future research:

“To ensure widespread clinical adoption, future efforts could expand patient cohorts and incorporate different PD phenotypes, medication regimens, and disease stages. Larger longitudinal studies can validate the sensitivity of these vocal biomarkers for detecting minimal or prodromal PD symptoms, as well as tracking advanced-stage patients. Moreover, combining speech metrics with other sensors (e.g., hand, head, and gait motion from mixed reality devices) may yield even more robust multi-modal markers, enhancing diagnostic power.”

Comments 3:

Medication status significantly influences all aspects of PD symptomatology. Understanding whether patients were evaluated in ON or OFF states and how levodopa may affect these speech parameters would improve clinical applicability.

Response 3: All patients included in our study were evaluated during their ON state. This was carefully documented in the Case Report Form (CRF), which recorded the exact time since the last levodopa dose. To ensure accuracy, we administered both the MDS-UPDRS-III and MoCA assessments immediately before the speech recordings. We have clarified this in the revised manuscript (Materials and Methods/Patient data, lines: 230-236; 246-256; 259-273), specifying the patients’ medication status and how it was controlled to ensure consistent clinical assessments.

Comments 4:

Conducting longitudinal assessments would be especially valuable in determining how these biomarkers correlate with disease progression, which is crucial for monitoring patients over time.

Response 4: We fully agree that longitudinal assessments are key to determining how these biomarkers evolve with disease progression. We are currently conducting a follow-up study that will track speech changes over time in the same cohort of PD patients. Our goal is to ascertain whether voice biomarkers can detect subtle changes associated with disease advancement and to confirm their utility as a monitoring tool. We look forward to sharing these future findings and discussing their implications for clinical practice.

Thank you once again for your thorough review and constructive feedback. We have taken your suggestions into careful consideration and have revised our manuscript to address each point. Your insights have strengthened our work’s clinical relevance, and we are excited to continue our investigations into how speech biomarkers can aid in the early detection and monitoring of Parkinson’s disease.

Kind regards,
Milosz Dudek

AGH University of Krakow
